# Home-based rehabilitation programme compared with traditional physiotherapy for patients at risk of poor outcome after knee arthroplasty: the CORKA randomised controlled trial

Karen L Barker [ORCID],[1] Jonathan Room,[2] Ruth Knight [ORCID],[3] Susan Dutton,[3] Francine Toye,[4] Jose Leal [ORCID],[5] Nicola Kenealy,[1] Michael Maia Schlüssel,[1] Gary Collins [ORCID],[6] David Beard,[1] Andrew James Price,[1] Martin Underwood [ORCID],[7] Avril Drummond [ORCID],[8] Sarah Lamb,[9,10] CORKA Trial group

For numbered affiliations see end of article.

**Correspondence to**
Dr Karen L Barker;
Karen.Barker@ouh.nhs.uk

## ABSTRACT

**Objectives** To evaluate whether a home-based rehabilitation programme for people assessed as being at risk of a poor outcome after knee arthroplasty offers superior outcomes to traditional outpatient physiotherapy.

**Design** A prospective, single-blind, two-arm randomised controlled superiority trial.

**Setting** 14 National Health Service physiotherapy departments in the UK.

**Participants** 621 participants identified at high risk of a poor outcome after knee arthroplasty using a bespoke screening tool.

**Interventions** A multicomponent home-based rehabilitation programme delivered by rehabilitation assistants with supervision from qualified therapists versus usual care outpatient physiotherapy.

**Main outcome measures** The primary outcome was the Late-Life Function and Disability Instrument (LLFDI) at 12 months. Secondary outcomes were the Oxford Knee Score (a disease-specific measure of function), Knee injury and Osteoarthritis Outcome Score Quality of Life subscale, Physical Activity Scale for the Elderly, 5 dimension, 5 level version of Euroqol (EQ-5D-5L) and physical function assessed using the Figure of 8 Walk test, 30 s Chair Stand Test and Single Leg Stance.

**Results** 621 participants were randomised between March 2015 and January 2018. 309 were assigned to CORKA (Community Rehabilitation after Knee Arthroplasty) home-based rehabilitation, receiving a median five treatment sessions (IQR 4–7). 312 were assigned to usual care, receiving a median 4 sessions (IQR 2–6). The primary outcome, LLFDI function total score at 12 months, was collected for 279 participants (89%) in the home-based CORKA group and 287 participants (92%) in the usual care group. No clinically or statistically significant difference was found between the groups (intention-to-treat adjusted difference=0.49 points; 95% CI −0.89 to 1.88; p=0.48). There were no statistically significant differences between the groups on any of the patient-reported or physical secondary outcome measures at 6 or 12 months.

### Strengths and limitations of this study

► Large sample size and low loss to follow-up which ensured sufficient numbers to make the study sufficiently powered to detect any between group differences.

► The trial was pragmatic and used an innovative workforce model that could readily translate into National Health Service practice within existing commissioning and financial constraints.

► The study offers an alternative service delivery model in line with the latest UK guidance from National Institute for Health and Clinical Excellence.

► Limitations are that the screening tool did not identify patients that were a good fit of risk of poor outcome for the intervention that we developed.

► Another limitation, common to all trials of post-operative physiotherapy after knee arthroplasty was the absence of a no physiotherapy treatment control group (as postdischarge physiotherapy is a core component of UK arthroplasty practice it is not practicable to recruit to a no treatment comparator especially for those identified as being at higher risk of a poor outcome.

There were 18 participants in the intervention group reporting a serious adverse event (5.8%), only one directly related to the intervention, all other adverse events recorded throughout the trial related to underlying chronic medical conditions.

**Conclusions** The CORKA intervention was not superior to usual care. The trial detected no significant differences, clinical or statistical, between the two groups on either primary or secondary outcomes. CORKA offers an evaluation of an intervention utilising a different service delivery model for this patient group.

**Trial registration number** ISRCTN13517704.

## INTRODUCTION

Knee arthroplasty (KA) operations in the UK continue to rise with the National Joint Registry reporting over 100 000 in 2017 alone.[1] This is an increase of over 3% from 2010. The population receiving surgery is getting older, frailer and with more coexisting health conditions. Data from patient-reported outcome measures (PROMs) have shown that most patients achieve a satisfactory outcome. However, between 10% and 15% are unsatisfied or report little or no improvement after surgery.[2] We do not yet know how to identify these patients or target rehabilitation to improve their outcomes.

Physiotherapy provision in the UK varies following discharge from hospital after KA but typically patients will continue to have rehabilitation needs to help restore muscle strength and endurance, range of motion, walking distance and performance of higher-level functional activities. Patients are commonly referred for further physiotherapy in an outpatient setting to assist with these issues. The timing of this referral varies, with some hospitals advocating immediate commencement of therapy after discharge from hospital and others waiting until the first postoperative review at around 6 weeks before considering referral. This typically involves between 4 and 6 sessions of physiotherapy in an outpatient clinic setting. Patients do not routinely receive occupational therapy input following knee arthroplasty surgery unless they are highlighted to have particular difficulties with their daily activities or have specific barriers to discharge home. Concern has been raised that many exercise programmes lack adequate intensity to enable optimal recovery.[3 4] Internationally, where much longer courses of physiotherapy are often provided, research has indicated that 12–18 hours of physiotherapy[5] or a mean of 17 visits[6] may be needed to produce a benefit. These levels of care are substantially higher than those provided in the UK and, in the current economic climate, may be more than the National Health Service (NHS) can afford.

Given the rising number of operations, the relatively limited therapy resources available, and the increasing age and frailty of patients receiving this surgery, it is important that rehabilitation resources are concentrated on those patients who need the most help to achieve a good outcome. Not all patients need postoperative outpatient physiotherapy and some will make a full recovery using self-directed rehabilitation.[7–10] Current rehabilitation strategies do not meet the needs of all patients, particularly those who are socially isolated, do not have easy access to transport, and are frail.[11] It is thus pertinent to focus our resources on those who are most likely to be at risk of a poor outcome, least likely to be able to engage with a self-management approach, and able to benefit most from rehabilitation input To maximise access to therapy and to provide this within a similar cost envelope to current practice we designed an intervention to be delivered in the patients' homes utilising a workforce model that use rehabilitation assistants supported by qualified therapy staff, drawing on the success of this model in delivering therapy programmes such as the Otago falls prevention intervention.[12]

This trial was carried out to determine if a targeted home-based intervention for those at risk of poor outcome, delivered by a mix of qualified staff and rehabilitation assistants, offers superior outcomes in patient reported and physical function at 1 year following KA compared with traditional outpatient physiotherapy.

## METHODS

The CORKA trial was a multicentre prospective, single-blinded, two-arm randomised controlled superiority trial with assessment for the clinical outcomes at baseline, 6 and 12 months. The study was carried out according to the published protocol.[13]

### Recruitment

Participants were recruited from 14 NHS Trusts across England. Patients aged 55 years and over due to undergo knee arthroplasty were sent information concerning the trial prior to their preoperative assessment appointment. Those expressing interest were screened in the clinic using a screening tool developed for the trial. This aimed to identify patients at risk of a poorer outcome based on data collected as part of standard pre-operative assessment screening. The tool was developed using all data from the Knee Arthroplasty Trial dataset and further details of its development has been previously published.[14] The screening tool had a range of possible scores from 0 to 10, patients with a score of 5 or more were classified as being at increased risk of a poor outcome after 1 year defined as an Oxford Knee Score (OKS) of 26 or less (online supplemental appendix A). The performance of the model was fair, as measured by its discrimination. Participants who were eligible were screened again post-operatively to confirm their eligibility and consent. All participants provided written informed consent. Exclusion criteria included any major perioperative complications, further surgery planned within the next 12 months and absolute contraindications to exercise. Screening was performed by research clinicians at the local sites, and screening logs recorded the reasons for ineligibility and why eligible patients declined to participate.

### Randomisation

Randomisation took place on the third postoperative day, or on hospital discharge if earlier. Should any participant have experienced a severe medical perioperative event they would not be entered into the trial but continue with usual care as able. Participants were randomly allocated by a web-based randomisation system to either 'usual care' or 'home-based rehabilitation programme' in a 1:1 ratio using permuted blocks of various sizes (sizes 2, 4 and 6 in a 1:2:1 ratio, that is on average there were twice as many blocks of size 4 than either size 2 or size 6) to prevent prediction of treatment allocation, and this was stratified by recruitment site. Once randomised,

participants and those delivering the rehabilitation were aware of the treatment allocation due to the nature of the intervention. Those carrying out follow-up outcome measurements remained blinded to treatment allocation.

## Interventions

### CORKA home-based intervention

The CORKA intervention started up to 4 weeks after surgery, with most participants seen within 2 weeks. In the period between surgery and the initial appointment participants continued with a home exercise programme given to them by ward physiotherapists in line with standard current practice.

The CORKA home-based intervention is a multicomponent rehabilitation programme developed in collaboration with clinicians and patients; it has been described in detail in a previous publication.[15] It consisted of an initial assessment appointment and up to six follow-up sessions. Its aim was to improve the function and participation in activities of participants at risk of a poor outcome after surgery. The primary component of the rehabilitation programme was an individually adapted exercise package conducted in participants' own homes. During the participants' initial assessment, the physiotherapist focused on tasks that were identified as being problematic. The exercise programme was then tailored to the individuals' needs and goals. If an exercise was felt to be unsafe for any particular participant, therapists could omit it and substitute with an additional exercise from the package; or use a more stable starting position. Additional components consisted of functional task practice, adherence approaches such as utilisation of techniques such as goal setting and exercise dairies. If required, they were provided with appropriate aids and equipment,

The physiotherapist undertook the initial assessment and prescription of the exercise programme, observed by the rehabilitation assistant. For future sessions, the rehabilitation assistant continued the programme and modified as required using treatment algorithms and decision aids to progress. The physiotherapist undertook one further session in the middle of the programme to review the participant's progress and their exercise programme. The way this was operationalised is outlined in online supplemental appendix B.

All therapists delivering the CORKA intervention received a training session of 2–3 hours, which included instructions on how to assess and treat CORKA participants, prescribe and progress the different categories within the exercise programme.

### Usual care

Those allocated to the usual care arm received standard postoperative physiotherapy. Usual care after surgery could vary considerably across the trial's different UK locations. However, it was highly likely that usual care would include several of the following: between one and six sessions of physiotherapy in an outpatient setting, class-based setting or hydrotherapy; written advice on

home exercises on discharge from hospital; and an assessment of any potential home requirements or barriers to discharge by an occupational therapist. To standardise usual care as much as possible, participants were expected to attend a minimum of 1 session and a maximum of six sessions of usual care physiotherapy.

Quality assurance checks took place at all CORKA research sites. They included fidelity checks during which assessment and treatment delivery were observed. All aspects of the intervention were checked against a predefined fidelity checklist created in line with the study protocol.

## Outcomes

Baseline data were collected face to face, no more than 4 weeks before surgery. Follow-up data collection was by face-to-face clinical assessments at 6 and 12 months following randomisation. Where face-to-face assessment was not possible, postal and telephone data collection methods were used to obtain self-reported core data. Baseline data included the Functional Comorbidities index to provide baseline characteristics of the participants.

### Primary outcome

The primary outcome was the Late-Life Function and Disability Instrument (LLFDI) overall function score. It was developed specifically to assess change in community-dwelling older adults. It assesses and responds to meaningful change in two distinct outcomes: a person's ability to do discrete actions or activities using a 32-item function component (primary outcome) and a person's performance of socially defined life tasks using a 16-item disability component (secondary outcome).

### Secondary outcomes

Secondary outcomes consisted of self-reported and physical measures. The self-reported measures were the OKS, Physical Activity Scale for the Elderly (PASE), Knee Injury and Osteoarthritis Outcome Score (KOOS) Quality of Life subscale and EQ-5D-5L questionnaires. The physical measures were the Figure of 8 Walk Test (F8WT), 30 s Chair Stand Test (30SCST) and Single Leg Stance (SLS). A health resource diary collected the exercises undertaken, medication taken, use of healthcare services and personnel, and falls.

## Analysis

A detailed statistical analysis plan has previously been published.[16] The LLFDI score was chosen as a clinically relevant outcome for the population and the sample size calculation was therefore based on this variable. As there is no formal minimally clinically important difference for the LLFDI the calculation was based on a moderately small standard effect size of 0.275 which gave a sample of 620 participants (310 per arm) with 90% power, 5% (two-sided) significance, allowing for 10% lost to follow-up.

Two analysis populations were considered: the intention-to-treat (ITT) population and the per-protocol (PP) population. The ITT population included all

randomised participants analysed according to their allocated intervention. The PP population included only participants who received at least one session of their allocated intervention, did not receive more treatment than intended (more than six sessions of usual care or seven sessions of home-based rehabilitation) and provided follow-up data.

LLFDI function scores at 6 and 12 months postrandomisation were summarised by treatment group and analysed using a linear mixed effects model with repeated measures adjusted for baseline score and recruiting site (stratification factor). Time was treated as categorical and an interaction between the outcome measurement time point and randomised group was included to allow the treatment effect to be estimated at each time point, reported as the adjusted mean difference in LLFDI between groups with 95% CI and associated p value. The underlying assumptions of this model were assessed. A 12-month postrandomisation was considered the primary endpoint. The primary analysis was performed for the ITT population using multiple imputation (MI) to impute missing data. MI was performed separately for each treatment group and the model included type of knee replacement (Total -TKR, Unicompartmental knee replacement UKR), recruiting centre, gender, whether or not the participant had had previous lower limb surgery, Charnley classification score and current mobility support used. As sensitivity analyses this was repeated for the ITT population using available cases, the PP population using available cases and using a complier average causal effect (CACE) analysis.[17]

Linear mixed effects models with repeated measures, similar to that described for the primary outcome, were used to analyse each of the secondary PROMs (LLFDI disability limitation, LLFDI disability frequency, OKS, KOOS QoL, PASE, EQ-5D- 5L utility and VAS), 30SCST (number of stands) and F8WT (time and steps). Differences between the two groups in SLS times were compared using a Wilcoxon rank-sum test since the assumptions of the linear mixed effects model were not satisfied. All of these analyses were performed for the ITT population using available cases only. For the key secondary outcomes (LLFDI disability limitation, LLFDI disability frequency, OKS and KOOS QoL), these analyses were repeated for the ITT population using MI (imputation model as outlined for the primary outcome) and the PP population using available cases.

The number and proportion of serious adverse events (SAEs) were reported by treatment group, as well as the number and proportion of participants experiencing an SAE. The rates were compared across treatment groups.

### Additional analysis

An economic analysis and nested qualitative analysis were conducted as part of this study. Both economic and qualitative analysis will be reported elsewhere.

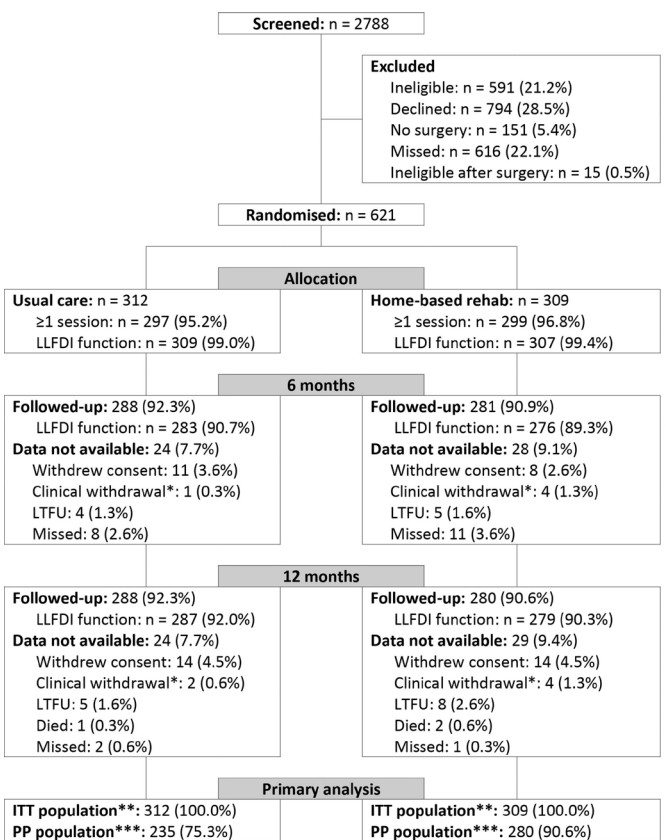

**Figure 1** CONSORT flow chart. CONSORT, Consolidated Standards of Reporting Trials.

### Patient and public involvement

The study was developed with a group of service users that identified concern about the access to rehabilitation for older more frail patients after knee arthroplasty. The intervention was developed with active involvement of patient and public involvement (PPI) stakeholders who worked alongside us to develop the intervention. A PPI representative collaborated on the application for funding and was a full voting member of the trial steering group which benefitted from their expertise and active engagement throughout the period of the study.

### RESULTS

The CORKA recruited participants between March 2015 and January 2018. A total of 2788 patients were screened for eligibility and 621 participants were randomised, 312 to usual care and 309 to the home based rehabilitation programme. A CONSORT flow chart summarises the flow of participants through the trial including details on the number of participants randomised and the numbers allocated to and receiving at least 1 session of each treatment (see figure 1). Of the 621 participants, 34 (5.5%) withdrew from follow-up during the course of the trial.

**Table 1** Descriptive characteristics at baseline by treatment group

| | Usual care (n=312) | Home-based rehab (n=309) | Total (n=621) |
|---|---|---|---|
| Gender* | | | |
| Male | 125 (40.1%) | 125 (40.5%) | 250 (40.3%) |
| Female | 187 (59.9%) | 184 (59.5%) | 371 (59.7%) |
| Age (years)† | 70.18 (8.14) | 70.67 (8.01) | 70.42 (8.07) |
| BMI† | 31.65 (4.99) | 31.34 (4.48) | 31.50 (4.74) |
| Side of operation* | | | |
| Right | 169 (54.2%) | 169 (54.7%) | 338 (54.4%) |
| Left | 142 (45.5%) | 139 (45.0%) | 281 (45.2%) |
| Not recorded | 1 (0.3%) | 1 (0.3%) | 2 (0.3%) |
| Knee arthroplasty type* | | | |
| TKA | 229 (73.4%) | 231 (74.8%) | 460 (74.1%) |
| UKA | 82 (26.3%) | 77 (24.9%) | 159 (25.6%) |
| Not recorded | 1 (0.3%) | 1 (0.3%) | 2 (0.3%) |
| ASA grade* | | | |
| Healthy | 38 (12.2%) | 55 (17.8%) | 93 (15.0%) |
| Mild systemic disease | 218 (69.9%) | 202 (65.4%) | 420 (67.6%) |
| Severe systemic disease | 43 (13.8%) | 44 (14.2%) | 87 (14.0%) |
| Not recorded | 13 (4.2%) | 8 (2.6%) | 21 (3.4%) |
| Falls in the last year* | | | |
| Yes | 77 (24.7%) | 89 (28.8%) | 166 (26.7%) |
| No | 235 (75.3%) | 220 (71.2%) | 455 (73.3%) |
| If yes, no of falls‡ | 1 (1, 3) | 2 (1, 3) | 2 (1, 3) |
| Previous lower limb surgery* | | | |
| Yes | 200 (64.1%) | 189 (61.2%) | 389 (62.6%) |
| No | 112 (35.9%) | 120 (38.8%) | 232 (37.4%) |
| Screening tool score‡ | 6 (5, 6) | 6 (5, 6) | 6 (5, 6) |
| Charnley ABC* | | | |
| A—single knee arthroplasty | 134 (42.9%) | 138 (44.7%) | 272 (43.8%) |
| B—both knees affected | 140 (44.9%) | 145 (46.9%) | 285 (45.9%) |
| C—multiple joint disease/other disability | 38 (12.2%) | 26 (8.4%) | 64 (10.3%) |
| Stairs mobility* | | | |
| Normal | 19 (6.1%) | 19 (6.1%) | 38 (6.1%) |
| One step at a time | 34 (10.9%) | 39 (12.6%) | 73 (11.8%) |
| Down with rail | 18 (5.8%) | 19 (6.1%) | 37 (6.0%) |
| Up/down with rail | 225 (72.1%) | 216 (69.9%) | 441 (71.0%) |
| Unable down | 2 (0.6%) | 3 (1.0%) | 5 (0.8%) |
| Unable | 14 (4.5%) | 12 (3.9%) | 26 (4.2%) |
| Missing | 0 (0.0%) | 1 (0.3%) | 1 (0.2%) |
| Support mobility* | | | |
| None | 178 (57.1%) | 178 (57.6%) | 356 (57.3%) |
| Stick outdoors | 83 (26.6%) | 78 (25.2%) | 161 (25.9%) |
| Stick always | 34 (10.9%) | 31 (10.0%) | 65 (10.5%) |
| Two sticks | 6 (1.9%) | 7 (2.3%) | 13 (2.1%) |
| Two crutches | 5 (1.6%) | 7 (2.3%) | 12 (1.9%) |
| Walking frame | 6 (1.9%) | 8 (2.6%) | 14 (2.3%) |
| Functional Comorbidity Index§ | | | |
| 0 | 189 (60.6%) | 176 (57.0%) | 365 (58.8%) |
| 1–3 | 112 (35.9%) | 125 (40.5%) | 237 (38.2%) |

Continued

**Table 1** Continued

|  | Usual care (n=312) | Home-based rehab (n=309) | Total (n=621) |
|---|---|---|---|
| 4–6 | 8 (2.6%) | 6 (1.9%) | 14 (2.3%) |
| 7+ | 2 (0.6%) | 0 (0.0%) | 2 (0.3%) |
| Missing | 1 (0.3%) | 2 (0.6%) | 3 (0.5%) |

*These values are reported as numbers and percentages.
†These values are reported as means and SD.
‡These values are reported as medians and IQR.
§The Functional Comorbidity Index (85) counts the number of comorbidities experienced by each patient, giving more weight to more severe conditions. This was categorised in four groups: (1) no comorbidities (0), (2) small number of comorbidities,[1–3] (3) medium number of comorbidities[4–6] and (4) high number of comorbidities (7+).
ASA, American Society of Anesthesiologists; BMI, body mass index; TKA, total knee arthroplasty; UKA, unicompartmental knee arthroplasty.

Three participants died between the 6 and 12 months follow ups.

The mean age of participants was 70.4 (SD 8.07), 59.7% were female and mean BMI was 31.5 kg/m² (SD 4.74). Most participants scored 5 or 6 on the screening tool (79.5%) with 5 being the minimum score needed to be eligible for the study and indicating a risk of poor outcome. The baseline characteristics of the participants is summarised in table 1.

Participants were defined as complying with usual care if they attended at least one treatment session and for home-based rehabilitation at least four treatment sessions. We used different thresholds for the two treatments as usual care varied considerably between sites including as little as one session. Therefore, compliance in this arm was defined as participants receiving at least one session. For the intervention arm, it was prespecified that at least four of the seven planned sessions were required to be compliant. Therapy compliance was 95.2% in usual care and 87.1% for the CORKA intervention. The median number of sessions for usual care was 4 (IQR 2–6) and for the CORKA intervention was 5 (IQR 4–7).

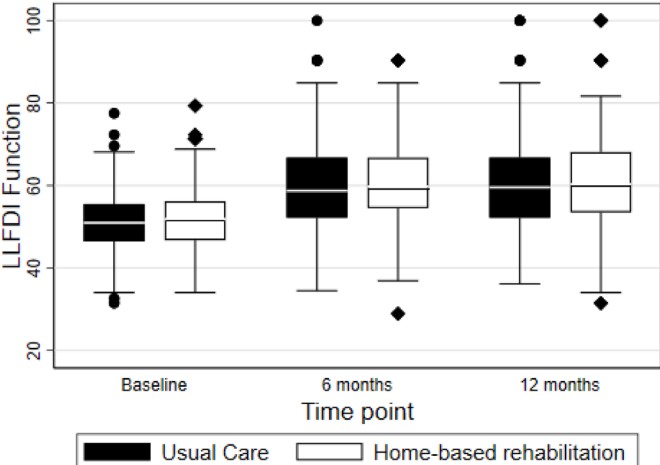

**Figure 2** The LLFDI function score from baseline to 12 months postrandomisation for each treatment group. LLFDI, Late-Life Function and Disability Instrument.

### Primary outcome analysis

For the ITT analysis using MI there was no statistically significant difference between the two treatment groups at the primary time point of 12 months postrandomisation (adjusted difference 0.49, 95% CI −0.89 to 1.88) or at 6 months (adjusted difference 0.66, 95% CI −0.70 to 2.03). This equates to a standardised effect size of 0.03 at 12 months.

Trends in LLFDI function scores over time are plotted for each treatment group in figure 2. There was a substantial improvement between baseline and 6 months in both groups, with minimal additional difference between 6 and 12 months postrandomisation.

None of the sensitivity analyses conducted (ITT using available cases, PP analysis, CACE analysis) demonstrated a different result.

### Secondary outcome analysis

For the secondary PROMS ITT analysis using available cases, no significant differences between the two treatment groups at either time point (6 and 12 months) was identified for any of the secondary outcomes (see table 2). The sensitivity analyses of the key secondary PROMS identified only one difference, of borderline statistical significance, for the LLFDI disability limitation at 6 months using the ITT population and MI (adjusted difference 2.67, 95% CI 0.14 to 5.19).

The physical measures are summarised by treatment group over time in table 3. With the exception of one of the SLS tests, there were no differences in change in performance between the groups on the physical measures.

The total number of SAEs experienced in this trial was low; 18 events in 14 participants in the usual care group (4.5%) and 20 events in 18 participants in the CORKA intervention group (5.8%), experienced. Adverse events were independently reviewed and categorised using the System Organ Classes codes and as related to the intervention or not. For all SAEs, only one was classified as 'possibly related'. In total, there were 7 SAEs in the musculoskeletal and connective tissue disorder category, with one categorised as related to the treatment interventions (table 4). The majority of the SAEs were related to

**Table 2** Comparison of treatment groups for secondary outcomes (intention-to-treat population using available cases)

| | | Usual care | | Home-based rehab | | | |
| | | Mean (SD) | Total | Mean (SD) | Total | Adjusted difference (95% CI) | P value |
|---|---|---|---|---|---|---|---|
| LLFDI disability (frequency) | Baseline | 51.28 (7.25) | 311 | 51.55 (7.46) | 308 | – | – |
| | 6 months | 54.27 (6.86) | 283 | 54.91 (6.85) | 275 | 0.64 (–0.50 to 1.78) | 0.27 |
| | 12 months | 54.13 (6.87) | 287 | 55.05 (6.87) | 279 | 0.93 (–0.21 to 2.06) | 0.11 |
| LLFDI disability (limitation) | Baseline | 66.26 (11.96) | 312 | 66.25 (12.48) | 305 | – | – |
| | 6 months | 72.33 (15.25) | 284 | 74.72 (15.24) | 275 | 2.39 (–0.14 to 4.92) | 0.06 |
| | 12 months | 74.10 (15.27) | 287 | 74.75 (15.28) | 277 | 0.65 (–1.87 to 3.18) | 0.61 |
| OKS | Baseline | 20.59 (7.50) | 312 | 20.81 (7.31) | 308 | – | – |
| | 6 months | 35.69 (7.69) | 284 | 36.23 (7.69) | 277 | 0.54 0.73 to 1.81) | 0.40 |
| | 12 months | 37.34 (7.71) | 287 | 37.80 (7.71) | 278 | 0.46 (–0.81 to 1.73) | 0.48 |
| KOOS – QoL subscale | Baseline | 25.50 (16.58) | 301 | 25.50 (15.89) | 300 | – | – |
| | 6 months | 61.26 (22.31) | 273 | 62.47 (22.29) | 269 | 1.21 (–2.55 to 4.97) | 0.53 |
| | 12 months | 65.53 (22.37) | 277 | 65.26 (22.36) | 271 | –0.27 (–4.02 to 3.47) | 0.89 |
| PASE | Baseline | 121.24 (73.90) | 302 | 115.11 (72.95) | 297 | – | – |
| | 6 months | 150.62 (66.72) | 259 | 149.41 (66.46) | 245 | –1.22 (–12.86 to 10.42) | 0.84 |
| | 12 months | 156.58 (66.82) | 263 | 154.63 (66.73) | 251 | –1.95 (–13.51 to 9.61) | 0.74 |
| EQ5D-5L utility | Baseline | 0.52 (0.22) | 308 | 0.52 (0.23) | 308 | – | – |
| | 6 months | 0.75 (0.18) | 277 | 0.75 (0.18) | 275 | 0.00 (–0.03 to 0.03) | 0.85 |
| | 12 months | 0.76 (0.18) | 282 | 0.76 (0.18) | 278 | –0.00 (–0.03 to 0.03) | 0.99 |
| EQ5D-5L VAS | Baseline | 68.53 (19.46) | 312 | 68.03 (19.13) | 308 | – | – |
| | 6 months | 75.81 (14.55) | 283 | 75.51 (14.54) | 275 | –0.30 (–2.71 to 2.11) | 0.81 |
| | 12 months | 76.80 (14.57) | 287 | 77.05 (14.58) | 278 | 0.24 (–2.16 to 2.65) | 0.84 |

Totals refer to the total number of participants in each group that provided each score at each time point. Adjusted differences were calculated using linear mixed effects models adjusted for stratification factors and baseline scores.
Each LLFDI disability scale ranges from 0 to 100 with higher scores indicating less disability.
Scores range from 0 to 48 with higher scores indicating better knee function.
Scores range from 0 to 100 with higher scores indicating better QoL. PASE, higher scores indicate greater levels of physical activity.
EQ5D-5L utility scores range from –0.594 to 1 and EQ5D-5L VAS ranges from 0 to 100, in both cases higher scores indicate better QoL.
KOOS, Knee injury and Osteoarthritis Outcome Score; LLFDI, Late-Life Function and Disability Instrument; OKS, Oxford Knee Score; PASE, Physical Activity Scale for the Elderly; QoL, quality of life; VAS, Visual Analogue Scale.

medical conditions unconnected with the participant's surgery and were classified into categories such as cardiac disorders, renal and urinary disorder or immune system disorders. There was not a significant difference in the number of participants experiencing an SAE between the two groups (risk difference 1.3%, 95% CI –2.1% to 4.8%).

## DISCUSSION

No statistical or clinically meaningful differences were found between CORKA and usual outpatient physiotherapy. Outcomes across a range of measures were closely matched irrespective of which of the two interventions was received. The use of a workforce model using rehabilitation assistants offered flexibility of staffing and did not result in any increase in adverse events.

### Strengths and limitations of the study

Strengths of this study include the large sample size and low lost to follow-up which ensured sufficient numbers to make the study sufficiently powered to detect any between group differences. The trial was pragmatic and utilised an innovative workforce model that could readily translate into NHS practice within existing commissioning and financial constraints. The study offers an alternative service delivery model in line with the latest UK guidance from National Institute for Health and Clinical Excellence (NICE).

Limitations are that the screening tool did not identify patients that were a good fit of risk of poor outcome for the intervention that we developed. The screening tool has not been validated; but uses items that are used by many knee arthroplasty trials to determine entry, these are broad and inclusive, reflecting common clinical practice. Recent recommendations from the UK NICE noted that it would be useful if a tool existed that indicated those who would benefit from supervised rehabilitation and recognised that there is a significant proportion of patients to whom this will apply.[18] In developing our screening tool for the trial, we attempted just this, but recognise that there was a poor match between the patients screened and the intervention that we developed. Many of our participants were borderline high risk scoring just above the cut-point of 5, resulting in fewer participants at higher risk than anticipated, which may have affected the results.

**Table 3** Comparison of physical measures between treatment groups at follow-up time points

| | Usual care (n=312) | | Home-based rehab (n=309) | | | |
| --- | --- | --- | --- | --- | --- | --- |
| | Summary | N | Summary | N | Adjusted difference (95% CI) | P value |
| **30SCST # stands*** | | | | | | |
| Baseline | 8.2 (3.5) | 312 | 8.4 (3.7) | 309 | – | – |
| 6 months | 11.1 (3.1) | 278 | 11.0 (3.1) | 275 | −0.06 (-0.548 to 0.46) | 0.82 |
| 12 months | 11.7 (3.1) | 279 | 11.5 (3.1) | 276 | −0.22 (-0.74 to 0.3) | 0.41 |
| **F8WT (seconds)*** | | | | | | |
| Baseline | 11.8 (5.4) | 312 | 12.1 (4.9) | 309 | – | – |
| 6 months | 10.2 (3.6) | 277 | 9.6 (3.6) | 275 | −0.57 (-1.17 to 0.04) | 0.07 |
| 12 months | 9.3 (3.6) | 280 | 9.1 (3.6) | 274 | −0.17 (-0.78 to 0.43) | 0.58 |
| **F8WT steps*** | | | | | | |
| Baseline | 16.4 (4.5) | 312 | 17.3 (5.0) | 309 | – | – |
| 6 months | 15.5 (3.2) | 277 | 15.2 (3.2) | 275 | −0.36 (-0.91 to 0.18) | 0.19 |
| 12 months | 15.2 (3.2) | 280 | 14.9 (3.3) | 274 | −0.23 (-0.77 to 0.31) | 0.40 |
| **KA leg SLS average (seconds)*** | | | | | | |
| Baseline | 5.3 (2.6, 13.0) | 310 | 4.5 (1.9, 14.5) | 308 | – | – |
| 6 months | 10.0 (3.9, 30.3) | 279 | 10.0 (3.6, 26.8) | 276 | – | 0.57 |
| 12 months | 13.3 (5.9, 31.1) | 280 | 11.7 (3.4, 31.1) | 276 | – | 0.14 |
| **Contralateral leg SLS average† (seconds)** | | | | | | |
| Baseline | 7.2 (3.6, 22.0) | 310 | 6.3 (2.6, 18.0) | 308 | – | – |
| 6 months | 10.9 (4.2, 30.8) | 279 | 9.8 (3.5, 29.7) | 276 | – | 0.43 |
| 12 months | 14.7 (5.6, 30.7) | 280 | 11.0 (3.2, 29.7) | 276 | – | 0.03 |

Totals refer to the total number of participants in each group that provided each score at each time point. 30SCST, which records the number of stands in 30 s and where higher scores indicate better function. F8WT, where a quicker time and fewer steps indicates better function. SLS, in which stance time is averaged over three trials at standing on the same leg, with a maximum score of 45 s and higher scores indicate better function
*Summaries are mean (SD).
†Summaries are median (IQR).
F8WT, Figure of 8 Walk Test; 30SCST, 30 s Chair Stand Test; SLS, Single Leg Stance.

Another limitation, common to all trials of post-operative physiotherapy after knee arthroplasty, was the absence of a control group. As outpatient physiotherapy

**Table 4** SOC codes of SAEs by treatment group

| | Usual care (n=14; 4.5%) | Home-based rehab (n=18; 5.8%) |
| --- | --- | --- |
| Blood/lymphatic | 2 | 2 |
| Cardiac | 2 | 1 |
| Endocrine | 1 | 0 |
| Gastrointestinal | 2 | 2 |
| Immune system | 0 | 1 |
| Infections/infestations | 1 | 3 |
| Musculoskeletal | 1 | 6 |
| Nervous system | 0 | 1 |
| Renal/urinary | 2 | 0 |
| Respiratory/thoracic | 0 | 1 |
| Skin | 1 | 0 |
| Social circumstances | 1 | 0 |
| Vascular | 1 | 0 |
| Unknown | 0 | 1 |

SAEs, serious adverse events; SOC, System Organ Classes.

remains an accepted and expected core component of UK arthroplasty practice it was not deemed ethical or practicable to attempt to recruit to a no treatment comparator in a group who had already been identified as at higher risk of a poor outcome. As with most physiotherapy trials, it was not feasible to blind patients or therapists to treatment allocation. We also chose to deliver an intervention that met current commissioning guidance of around six sessions. It is arguable that more intensive treatment as is common in other countries may have been more effective, but this is not supported by studies using longer interventions.[19–24]

We chose to use the LLFDI as our primary outcome over a more knee focused outcome as it is designed to map to the WHO ICF and assess change in community-dwelling older adults encompassing the range of activities relevant for older, potentially frailer patients after arthroplasty. At the time of designing the trial the minimal clinically important difference (MCID) was not published. Subsequently, Beauchamp et al reported that substantial changes are reflected in a 5-point change in the overall function scale.[25] We found substantial changes from baseline to 1 year within both groups, but no between group effect. The CORKA home-based intervention showed an overall change of 8.9 points and usual care 8.7 points,

indicating that the combined effect of surgery and rehabilitation unsurprisingly produced a highly significant improvement in function. Most of this improvement was probably due to the surgical procedure, but we cannot disaggregate the contribution of physiotherapy to quantify the contribution of one without the other. We found no significant between group difference. It is arguable that a more specific outcome measure targeted at knee arthroplasty may have been a better choice. We included the OKS as a secondary measure to address this potential limitation. The results of the LLFDI and OKS were well matched.

### Implications for patients and policy

The benefit of postoperative physiotherapeutic interventions is poorly established. A recent evidence review by NICE, of all patients receiving arthroplasty surgery, found no clinically important differences in quality of life, PROMS or functional outcomes between group or individually based supervision and self-directed rehabilitation. They recommended self-directed rehabilitation as an acceptable option for people who are well enough and have personal circumstances that make it possible; but recognised that there are likely to be subgroups who will benefit from a more supervised rehabilitation programme. The CORKA trial is one of few trials to specifically target individuals who are at risk of poor outcome post surgery. The TRIO trial[26] also targeted physiotherapy at patients considered at risk of poor outcome, based on OKS at 6 weeks. Similar to this study they found no statistical differences between the two treatment arms of outpatient physiotherapy and a single physiotherapy review and home exercise based regimen.

Although we screened for patients deemed at risk of poor outcome the OKS of 37.8 points was higher than the reported average for all knee arthroplasty patients at 1 year of 36 points. The TRIO trial similarly found their results, although a little lower, were similar to the unselected average.[26] Most of this improvement was probably due to the surgical procedure, but we cannot disaggregate the contribution of physiotherapy to quantify the contribution of one without the other.

The CORKA intervention was based on preoperative outcome measures, with a home-based rehabilitation programme. There may also be benefits to home supervised rehabilitation that are not captured by the current evidence or the outcome measures used in existing studies. For example, participants in our embedded qualitative study were very positive about the CORKA home-based intervention and the supervision provided by the therapists and rehabilitation assistants.[27] The CORKA home-based intervention could also benefit, for example, those too frail to travel a significant distance to a larger hospital. Using the CORKA combination of qualified therapists and rehabilitation assistants might be a practical option for certain remote areas of the UK or countries that have large geographical areas not covered by therapists. This suggests that there is an argument for choice, according to personal preference or the practicalities of service provision.

It also addresses the workforce shortage issue by using an innovative workforce model of advanced rehabilitation assistants, moving UK service provision closer to that which has been proven to be effective in North America, where the use of physical therapy assistant graded staff is well embedded. This is a particular concern given both the projected increased need for joint arthroplasty over the next decade to accommodate an ageing population and the pressure of potential reductions in NHS funding. Evaluating the value of treatment modalities offered to these patients is crucial because many more patients are being discharged home earlier from the acute setting leaving less time available for acute physical recovery, rehabilitation and education in hospital, increasing the potential burden of care for these patients and their families.

### Future research

Further research should focus on developing a screening tool that is more sensitive in identifying those patients who will benefit from additional input beyond self-directed rehabilitation as recommended by the NICE guidance on primary joint replacement.[18]

The CORKA home-based intervention was delivered by rehabilitation assistants supervised by qualified therapists in a ratio of 5:2 sessions. Given the workforce challenges in the NHS further work looking at different workforce models and interventions solely using rehabilitation assistants should be explored.

### CONCLUSIONS

This large randomised controlled trial found no important differences in outcomes when rehabilitation was delivered using either a home based, rehabilitation assistant delivered rehabilitation package or a traditional outpatient physiotherapy model. Home-based rehabilitation was feasible but did not confer benefits over standard out patient physiotherapy.

**Author affiliations**
[1]Nuffield Department of Orthopaedics, Rheumatology and Musculoskeletal Sciences, University of Oxford, Oxford, UK
[2]Nuffield Orthopaedic Centre, Oxford University Hospitals NHS Foundation Trust, Oxford, UK
[3]Oxford Clinicial Trials Research Unit (OCTRU), Centre Statistics in Medicine, University of Oxford, Oxford, Oxfordshire, UK
[4]Nuffield Orthopaedic Centre Physiotherapy Research Unit, Oxford University Hospitals NHS Foundation Trust Nuffield Orthopaedic Centre, Oxford, UK
[5]Nuffield Department of Population Health, University of Oxford, Oxford, UK
[6]Centre for Statistics in Medicine, University of Oxford, Oxford, UK
[7]Warwick Medical School, Warwick University, Coventry, UK
[8]Div of Rehabilitation and Ageing, University of Nottingham, Nottingham, UK
[9]College of Medicine and Health, University of Exeter, Exeter, UK
[10]NDORMS, University of Oxford, Oxford, UK

**Acknowledgements** The authors would like to acknowledge the site principal investigators and research clinicians. Leon Palmer-Wilson, Ana Glennon (Ashford

Hospital), Helen Wilson (Countess of Chester Hospital), Christian Brookes, Denise Hill (Dorset County Hospital), Susan Dowdle, Hazel Burt (Dorset HealthCare Hospital), Jane Harrison (Epsom Hospital), Sarah Adcock, Justine Theaker (Manchester Royal Infirmary), Gladys Nadar Arulmani (Medway Community Hospital), Sunil Jain (Medway Maritime Hospital), Mike Reed (North Tyneside General Hospital), Jonathan Room (Oxford University Hospitals), Genevieve Simpson, Gemma Knight (Royal United Hospitals Bath), Tricia Monroe (Moorgreen Hospital), Gareth Stephens, Sarah Rich (Royal Orthopaedic Hospital Birmingham).

**Collaborators** CORKA Study teamLeon Palmer-Wilson, Ana Glennon (Ashford Hospital), Helen Wilson (Countess of Chester Hospital), Christian Brookes, Denise Hill (Dorset County Hospital), Susan Dowdle, Hazel Burt (Dorset HealthCare Hospital), Jane Harrison (Epsom Hospital), Sarah Adcock, Justine Theaker (Manchester Royal Infirmary), Gladys Nadar Arulmani (Medway Community Hospital), Sunil Jain (Medway Maritime Hospital), Mike Reed (North Tyneside General Hospital), Jonathan Room (Oxford University Hospitals), Genevieve Simpson, Gemma Knight (Royal United Hospitals Bath), Tricia Monroe (Moorgreen Hospital), Gareth Stephens, Sarah Rich (Royal Orthopaedic Hospital Birmingham).

**Contributors** KLB (professor of physiotherapy)—chief Investigator, led funding application, trial conception and design, development of interventions, supervision, and writing and reviewing report. JR (research physiotherapist)—principal investigator, development of intervention and writing and reviewing report. RK (statistician)—conducting statistical analysis and writing and reviewing report. SD (statistician)—designing and conducting statistical analysis, and writing and reviewing report. FT (qualitative lead)—conducting interviews and analysis, and writing chapter for report. JL (health economist)—designing and conducting economic analysis, supervision, and writing and reviewing report. NK (trial manager)—trial management and writing and reviewing report. MMS (statistician)—screening tool development and writing for report. GC (coapplicant)—supervised screening tool development. DB (coapplicant)—clinical trials expertise and contributed to report. AJP (coapplicant)—surgical expertise and contributed to report. MU (coapplicant)—primary care expertise and contributed to report. AD (coapplicant)—occupational therapy expertise and contributed to report. SL (coapplicant)—clinical trials expertise, key member of trial management group, and writing and reviewing report.

**Funding** This trial was funded by the National Institute for Health Research Health Technology Assessment programme (HTA 12/196/08).

**Competing interests** GC is a member of the HTA Commissioning Board. A Price has a consultancy with Zimmer Biomet, outside the submitted work. MU has grants from NIHR, during the conduct of the study; grants from NIHR, Personal fees from NIHR, Personal fees from NICE, non financial support from Stryker PLC, grants from SERCO. SL has grants from NIHR Health Technology Assessment Programme during the conduct of the study and is a member of the following Boards: HTA Additional Capacity Funding Board 2010-2015, HTA Clinical Trials Board 2010-2015, HTA End of Life Care and Add on Studies 2015-2015, HTA Funding Boards Policy Group (Formally CSG) 2010–2015, HTA MNCH Methods Group 2013–2015, HTA Post-board funding teleconference (PG members to attend) 2010–2015, HTA Primary Care Themed Call board 2013–2014, HTA Prioritisation Group 2014–2015, NIHR CTU Standing Advisory Committee 2012–2016.

**Patient consent for publication** Not required.

**Ethics approval** The study protocol was approved by the South-Central Research Ethics Committee (Reference 15/SC/0019). Ethics permission was obtained for all participating sites.

**Provenance and peer review** Not commissioned; externally peer reviewed.

**Data availability statement** Data are available upon reasonable request. Deidentified individual participant data will be available on reasonable request to the Chief Investigator. Data will be available beginning 9 months and ending 36 months following article publication to investigators who provide a methodologically sound proposal. Proposals may be submitted up to 36 months following article publication. To gain access data requestors will need to sign a data access agreement and the study in question.

**Open access** This is an open access article distributed in accordance with the Creative Commons Attribution 4.0 Unported (CC BY 4.0) license, which permits others to copy, redistribute, remix, transform and build upon this work for any purpose, provided the original work is properly cited, a link to the licence is given, and indication of whether changes were made. See: https://creativecommons.org/licenses/by/4.0/.

**ORCID iDs**
Karen L Barker http://orcid.org/0000-0001-9363-0383
Ruth Knight http://orcid.org/0000-0001-6810-2845
Jose Leal http://orcid.org/0000-0001-7870-6730
Gary Collins http://orcid.org/0000-0002-2772-2316
Martin Underwood http://orcid.org/0000-0002-0309-1708
Avril Drummond http://orcid.org/0000-0003-1220-8354

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
