## [Reviewer comments · BMJ Open]

ARTICLE DETAILS

TITLE (PROVISIONAL)	A home-based rehabilitation programme compared with traditional physiotherapy for patients at risk of poor outcome after knee arthroplasty: The CORKA randomised controlled trial.
AUTHORS	Barker, Karen L.; Room, Jonathan; Knight, Ruth; Dutton, Susan; Toye, Francine; Leal, Jose; Kenealy, Nicola; Maia Schlüssel, Michael; Collins, Gary; Beard, David; Price, Andrew; Underwood, Martin; Drummond, Avril; Lamb, Sarah

VERSION 1 – REVIEW

REVIEWER	de Sá Ferreira, Arthur Augusto Motta University Centre, Postgraduate Program in Rehabilitation Sciences
REVIEW RETURNED	10-Jun-2021

GENERAL COMMENTS	General comments This manuscript reports the results of a randomized clinical trial investigating whether a home-based rehabilitation program for people assessed as being at risk of a poor outcome after knee arthroplasty offers superior outcomes to traditional out-patient physiotherapy. The manuscript was submitted along with supplementary files including the assessment tool for risk stratification; full and illustrated reporting of the intervention; CONSORT 2010 and TIDieR checklists; full research protocol; and publications of the study protocol. In addition, the manuscript cites a preliminary validation of the risk assessment tool and the pre-planned statistical analysis. Overall, the manuscript is well-written and the research well-planned and conducted, and it seems the authors were able to provide accurate and transparent reporting of all stages of the trial so far. I have only minor suggestions for the authors to consider. Minor comments 1. Title. I acknowledge the complete study protocol includes cost-effectiveness assessment, but this manuscript does not report such analysis. Consider revising the title to better match the reported results.2. Introduction (lines 127-129). The study aim could explicitly mention the trial design as a 'superiority' of CORKA intervention compared to usual care as in Abstract (lines 27-29).3. Results (lines 275-276). Most participants were borderline high risk for poor outcome (scored 5 or 6 with being 5 the cutoff value).
---

	Consider discussing the possible impact of the dichotomization of risk for the inclusion of participants in the observed results. 4. Results (lines 286-287). It is apparent that this sentence is almost repeated from Analysis (lines 250-251), but no results are presented in this regard. 5. Discussion (line 329): Although I understand that costs of CORKA and usual care might differ, there are no new data in this manuscript to support this claim. Indeed, in the Introduction (lines 122-123) the authors state 'To maximise access to therapy and to provide this within a similar cost envelope to current practice'. 6. Discussion (line 387). A reference is indicated but not numbered. 7. Table 4. A possible typo in the usual care column (additional parenthesis) and empty cell value of 'unknown' under 'usual care'. 8. Figure 1. Please double-check the number of participants. In the 'usual care' arm, primary analysis, sample size (%) reads 235 (75.3%), which matches the number of participants that do not report a fall in the 'usual care' arm. However, Table 2 reports 251 to 279 participants depending on the secondary outcome. Finally, Table 3 reports 274 or 276 participants at 12 months depending on the physical measure. I assume these discrepancies are due to missing values but consider making them explicit in the text.
--	---

REVIEWER	Cameron, Claire University of Otago, Dunedin School of Medicine
REVIEW RETURNED	14-Jun-2021

GENERAL COMMENTS	This manuscript is a description of the results of a prospective, single-blind, two arm randomised controlled trial evaluating whether a home-based rehabilitation programme offers superior outcomes to traditional out-patient physiotherapy. I have been asked to provide a statistical review. Thank you for the opportunity. First of all, I appreciate the fact that this is a publication of a study that did not show any effect of the intervention. I think it very important, in general, that these 'null' results are published in a move against publication bias. I have read the statistical analysis plan and the protocol which provide a very clear description of the study and provide details on various aspects of the study that I was unable to glean from the manuscript that has been submitted. Unfortunately, the translation of this information into a coherent stand-alone manuscript reporting the results has not been successful. I have one statistical query regarding the study and the remaining comments are about the statistical aspects of this manuscript, in particular. I believe this paper needs a substantial rewrite to bring it up to the standard required for publication. My comments and queries: 1. The primary outcome that informed the sample size calculation was the LLFDI overall function score at 12 months. This means the effect of interest is the comparison of this outcome for the two arms of the trial at 12 months. However, the analysis plan describes using a linear mixed model to look at this primary outcome including an interaction term. This study has not been
--

powered to investigate an interaction. Also, the simple comparison of the outcome at 12 months does not seem to have been done. This relates to item 1 on “the ten most important and poorly reported CONSORT items as defined by a group of experts on the CONSORT statement” as provided to be to use as part of the review .

2. In the statistical plan and also in this manuscript, the authors refer to an ‘intention-to-treat population’ and a ‘per-protocol population’. These are not populations. They are methodological approaches for undertaking the analysis.

3. The authors list as a limitation, ‘the absence of a control group’. This is not correct, there is the intervention and a usual care group. The usual care group IS the control group. I would have thought it was standard practice to arrange trials this way and it surprises me that the authors do not see this as the control. If there actually was no control, this would be a fatal flaw in this study.

4. I am unsure of the description of the treatment groups. They seem to be better described elsewhere. I am also concerned that the usual care people are required to “attend a minimum of one session and a maximum of six sessions of usual care physiotherapy”. Is that still usual care?

5. In the manuscript the randomisation is poorly described. Looking at the analysis plan, I could understand what you were doing. You need to clarify this part – what does “permuted blocks of various sizes (sizes 2, 4 and 6 in a 1:2:1 ratio)” mean? (Item 3 of the list mentioned above)

6. Also, you haven’t really described how people are allocated to the groups. You have said that the participants and those delivering the rehabilitation were aware of the treatment allocation, but were they aware at the moment that the allocation took place? How did that happen? Was there any opportunity for participants to be moved to a different group because the people handing out the allocations thought an individual would be better off with the treatment? (Item 4 of the list)

7. The third and fourth paragraphs of the Analysis section were very unclear. This should be rewritten. The protocol and the statistical plan describe these parts very well.

8. I think there should be a justification of the use of multiple imputation (and what model of imputation you plan to use) when you have lost only 5.5% to followup.

9. I question the wisdom of replacing a linear mixed effects model with a Wilcoxon rank-sum test because a failure of the model assumptions.

10. There is a lot of talk of statistically significant differences in this manuscript – implying a commitment to p-values – when the CONSORT statement supports the reporting of effect sizes and confidence intervals (Item 6 on the list).

11. When you report results under the ‘Primary outcome analysis’ is this from your modelling or are you doing more simple tests here? I can’t tell.

12. There does not appear to be an economic or qualitative evaluation in here so those should really be taken out of the title.

13. The abundance of acronyms in the manuscript does not help this manuscript in terms of its readability.

These are my references:

Vadher, K., Knight, R., Barker, K. L., & Dutton, S. J. (2018). COMMUNITY-BASED REHABILITATION AFTER KNEE ARTHROPLASTY

	(CORKA): statistical analysis plan for a randomised controlled trial. Trials, 19(1), 1-7. Barker, K. L., Beard, D., Price, A., Toye, F., Underwood, M., Drummond, A., ... & Lamb, S. E. (2016). CCommunity-based Rehabilitation after Knee Arthroplasty (CORKA): study protocol for a randomised controlled trial. Trials, 17(1), 1-11. Hopewell, S., Boutron, I., Altman, D. G., Barbour, G., Moher, D., Montori, V., ... & Ravaud, P. (2016). Impact of a web-based tool (WebCONSORT) to improve the reporting of randomised trials: results of a randomised controlled trial. BMC medicine, 14(1), 1-17.
--	---

REVIEWER	Ganesh, Shankar Composite Regional Center for Persons with Disabilities, Physiotherapy
REVIEW RETURNED	22-Jun-2021

GENERAL COMMENTS	Dear authors, The results of the trial concluded that a custom-built community exercise program led by rehabilitation professionals and implemented by rehabilitation assistants is no better than outpatient rehabilitation in patients at risk of poorer outcomes following arthroplasty. The following points require clarification / rebuttal:  1. The status of the knee post-surgery (at the time of randomization) is not clear. 2. Evidence has shown that there is only a short-term benefit for out-patient physiotherapy and home-based exercises in improving physical function/pain following arthroplasty. Please state what specific points were considered in designing the exercise program for patients at risk of poorer outcomes. 3. Further, considering the limited effectiveness of out-patient physiotherapy in improving outcomes, do the authors believe it's important to have a screening tool with better discriminative validity? 4. Though the authors have provided a citation to the development of interventions for the CORKA home-based intervention, the role of the public-patient contribution could be added in the discussion section. Other points to consider:  1. The manuscript has considered a lot of points: evaluating 2 sets of interventions, discriminative validity for a screening test, and development of an alternative workforce model. The background may be modified further to justify the objectives of the study. 2. Please provide a citation to the point 'social isolation, lack of access to transport, and frail patients are risk factors for poor outcomes following knee arthroplasty. 3. When was the study undertaken? As the randomization was done after the surgery, what criteria were set by the authors to check the participants' post-surgery status? 4. What were the instructions provided to the professionals and assistants who participated in the study? How did the authors ensure the adherence between various groups that provided the interventions? 5. Please explain what the authors mean by adherence approaches 6. Was there a role of any confounding factors that might have influenced the results?
---

	7. There is not much discussion regarding the specific exercise program developed. Do the authors think there is a necessity to rework the exercise guidelines? 8. What do the authors mean by 'benefit from additional input'? Regards.
--	---

VERSION 1 – AUTHOR RESPONSE

From Reviewer 1		
This manuscript reports the results of a randomized clinical trial investigating whether a home-based rehabilitation program for people assessed as being at risk of a poor outcome after knee arthroplasty offers superior outcomes to traditional out-patient physiotherapy. The manuscript was submitted along with supplementary files including the assessment tool for risk stratification; full and illustrated reporting of the intervention; CONSORT 2010 and TIDieR checklists; full research protocol; and publications of the study protocol. In addition, the manuscript cites a preliminary validation of the risk assessment tool and the pre-planned statistical analysis. Overall, the manuscript is well-written and the research well-planned and conducted, and it seems the authors were able to provide accurate and transparent reporting of all stages of the trial so far. I have only minor suggestions for the authors to consider.	Thank you to the review for their comments on the conduct of the research.	
Title. I acknowledge the complete study protocol includes cost-effectiveness assessment, but this manuscript does not report such analysis. Consider revising the title to better match the reported results.	The title has been amended so that the cost-effectiveness (and qualitative) evaluations are not mentioned	Lines 1-2
Introduction (lines 127-129). The study aim could explicitly mention the trial design as a 'superiority' of CORKA intervention compared to usual care as in Abstract (lines 27-29).	Lines 127-129 have been edited so that it mirrors what is outlined in the abstract, including mention of 'superiority' In addition, the description of the trial design has been updated to include the designation as a superiority trial.	Lines 128-130 Lines 33 & 138
Results (lines 275-276). Most participants were borderline high risk for poor outcome (scored 5 or 6 with being 5 the cut-off value). Consider discussing the possible impact of the dichotomization of risk for the inclusion of participants in the observed results.	We have included discussion of this point in our Discussion	Lines 365-367.

Results (lines 286-287). It is apparent that this sentence is almost repeated from Analysis (lines 250-251), but no results are presented in this regard.	Thanks for pointing this out. We have removed this sentence and instead added one to the primary outcome analysis stating that none of the sensitivity analyses for this outcome (including CACE) demonstrated a different result.	Lines 301-302 and lines 316-317
Discussion (line 329): Although I understand that costs of CORKA and usual care might differ, there are no new data in this manuscript to support this claim. Indeed, in the Introduction (lines 122-123) the authors state 'To maximise access to therapy and to provide this within a similar cost envelope to current practice'.	This sentence has been removed.	L 347-8
Discussion (line 387). A reference is indicated but not numbered.	Corrected.	L 406
Table 4. A possible typo in the usual care column (additional parenthesis) and empty cell value of 'unknown' under 'usual care'.	The additional parentheses have been removed and semi-colons used to separate the numbers and percentages in the column headings. The correct number has been added to the empty cell	Table 4 – page 28
Figure 1. Please double-check the number of participants. In the 'usual care' arm, primary analysis, sample size (%) reads 235 (75.3%), which matches the number of participants that do not report a fall in the 'usual care' arm. However, Table 2 reports 251 to 279 participants depending on the secondary outcome. Finally, Table 3 reports 274 or 276 participants at 12 months depending on the physical measure. I assume these discrepancies are due to missing values but consider making them explicit in the text.	These numbers refer to the number of participants who contributed data to each analysis. The footnote to Figure 1 explains what the numbers included in the primary analysis refer to. The text has been updated to clarify that the secondary outcome analysis results are based on available cases	Line 321

	only. The footnotes to Tables 2 and 3 have also been updated to clarify which totals are reported.	
From Reviewer 2		
The primary outcome that informed the sample size calculation was the LLFDI overall function score at 12 months. This means the effect of interest is the comparison of this outcome for the two arms of the trial at 12 months. However, the analysis plan describes using a linear mixed model to look at this primary outcome including an interaction term. This study has not been powered to investigate an interaction. Also, the simple comparison of the outcome at 12 months does not seem to have been done. This relates to item 1 on “the ten most important and poorly reported CONSORT items as defined by a group of experts on the CONSORT statement” as provided to be to use as part of the review.	The linear mixed effects model was not used to test for an interaction but rather to make best use of all the available data (with participants able to contribute to the model even if they were followed up at only a single time point). We did not test for the significance of the interaction effect but rather used it to report treatment effects at each time point.	
In the statistical plan and also in this manuscript, the authors refer to an ‘intention-to-treat population’ and a ‘per-protocol population’. These are not populations. They are methodological approaches for undertaking the analysis. [EDITORS' NOTE: we are happy for the authors to rebut this comment, as this shorthand language is commonplace to describe trial analysis sets]	To refer to the sets of participants included in analyses in a trial as ‘populations’ and specifically as ‘intention-to-treat’ and ‘per-protocol’ is standard practice and therefore, whilst we acknowledge these are not populations in the strictest sense of the word, we have not changed this terminology.	
The authors list as a limitation, ‘the absence of a control group’. This is not correct, there is the intervention and a usual care group. The usual care group IS the control group. I would have thought it was standard practice to arrange trials this way and it surprises me that the authors do not see this	We have amended the wording to be clear that the limitation was the absence of a	L 84

as the control. If there was no control, this would be a fatal flaw in this study.	no treatment control group.	
I am unsure of the description of the treatment groups. They seem to be better described elsewhere. I am also concerned that the usual care people are required to “attend a minimum of one session and a maximum of six sessions of usual care physiotherapy”. Is that still usual care?	We have re-written this section to address. The usual care of ~6 sessions was selected after discussions with Clinical Commissioning groups and a survey of practice	L174-
In the manuscript the randomisation is poorly described. Looking at the analysis plan, I could understand what you were doing. You need to clarify this part – what does “permuted blocks of various sizes (sizes 2, 4 and 6 in a 1:2:1 ratio)” mean? (Item 3 of the list mentioned above)	Permuted blocks are lists of allocations of the stated length allocating the available treatments in the stated allocation ratio, for example for size 2 with 2 treatments in a 1:1 ratio there are 2 permutations: AB and BA. For larger blocks there are more permutations. The list of allocations was pre-generated before the trial and blocks in a 1:2:1 ratio means that blocks of size 4 were twice as likely to be used as those of sizes 2 or 6. We have clarified this in the text.	Lines 163-164
Also, you haven’t really described how people are allocated to the groups. You have said that the participants and those delivering the rehabilitation were aware of the treatment allocation, but were they aware at the moment that the allocation took place? How did that happen? Was there any opportunity for participants to be moved to a different group because the people handing out the allocations thought an individual would be better off with the treatment? (Item 4 of the list)	Participants were allocated using a web-based randomisation system. We have updated the text to reflect this.	Line 161 Lines 164-165

	Randomisation allocations were concealed up to the point of randomisation by using randomisation schedules embedded in an interactive web-based system. The person performing the allocation was informed of the treatment allocation the moment this took place using the web-based system described above this has been further clarified in the text.	
The third and fourth paragraphs of the Analysis section were very unclear. This should be rewritten. The protocol and the statistical plan describe these parts very well.	We have moved description of the CACE analysis of the primary outcome to the paragraph describing the analyses of this outcome. Additional details of the multiple imputation model used have been added.	Lines 249-250 Lines 245-248
I think there should be a justification of the use of multiple imputation (and what model of imputation you plan to use) when you have lost only 5.5% to follow up.	Multiple imputation was used in line with the pre-specified analysis plan. Sensitivity analyses were also performed using available cases only and did not demonstrate different results. Details of the multiple imputation model used have been included in the analysis methods section.	Lines 245-248

	Full details of the modelling approach are covered in the previously published statistical analysis plan (referenced in this paper).	
I question the wisdom of replacing a linear mixed effects model with a Wilcoxon rank-sum test because a failure of the model assumptions.	Whilst we agree that alternative approaches exist, this was in line with our pre-specified (and published) statistical analysis plan.	
There is a lot of talk of statistically significant differences in this manuscript – implying a commitment to p-values – when the CONSORT statement supports the reporting of effect sizes and confidence intervals (Item 6 on the list).	Results are presented as effects sizes and associated 95% confidence intervals along with associated p-values. With one exception all results in this trial did not reach the pre-specified threshold for statistical significance.	
When you report results under the ‘Primary outcome analysis’ is this from your modelling or are you doing more simple tests here? I can’t tell.	These results are based on the linear mixed effects model as outlined in the methods section. The footnote to Table 2 has been updated to clarify this.	Table 2 footnote
There does not appear to be an economic or qualitative evaluation in here so those should really be taken out of the title.	The title has been amended so that the cost-effectiveness (and qualitative) evaluations are not mentioned	Lines 1-2
The abundance of acronyms in the manuscript does not help this manuscript in terms of its readability.	We have reviewed and removed acronyms where appropriate	

From Reviewer 3		
The status of the knee post-surgery (at the time of randomization) is not clear.	Clarified in text	L173-176
Evidence has shown that there is only a short-term benefit for out-patient physiotherapy and home-based exercises in improving physical function/pain following arthroplasty. Please state what specific points were considered in designing the exercise program for patients at risk of poorer outcomes.	We emphasised functional activity practice, modifying exercises with a choice of starting positions of differing stability and the use of visual cues – please see Appendix and clarified text.	L173-199
Further, considering the limited effectiveness of out-patient physiotherapy in improving outcomes, do the authors believe it's important to have a screening tool with better discriminative validity?	Yes, in our discussion we have discussed the limitations of our screening tool and the numbers of participants who were just above the cut point. We have also discussed the recent NICE recommendation that such a tool be developed to identify patients who would benefit from supervised rehabilitation	L 357 L365-7 L361-364
Though the authors have provided a citation to the development of interventions for the CORKA home-based intervention, the role of the public-patient contribution could be added in the discussion section.	Details are included in our PPI section	L285-291
The manuscript has considered a lot of points: evaluating 2 sets of interventions, discriminative validity for a screening test, and development of an alternative workforce model. The background may be modified further to justify the objectives of the study.	Thank you for this observation. Within the limitations of the word count available we have tried to consider these in our introduction.	
Please provide a citation to the point 'social isolation, lack of access to transport, and frail patients are risk factors for poor outcomes following knee arthroplasty.	Reference added – Judge 2012	Line 123
When was the study undertaken? As the randomization was done after the surgery, what criteria were set by the authors to check the participants' post-surgery status?	Text added to randomisation section	Line162-163.
What were the instructions provided to the professionals and assistants who participated in the study? How did the authors	Additional text added	L202*204 L214-216

ensure the adherence between various groups that provided the interventions?		
Please explain what the authors mean by adherence approaches	Text added (brief due to space constraints, more detail in published intervention paper	L190-191
Was there a role of any confounding factors that might have influenced the results?	We attempted to control for confounding factors in the screening tool and stratification used to randomise	L 158
There is not much discussion regarding the specific exercise program developed. Do the authors think there is a necessity to rework the exercise guidelines?	No, we believe the NICE guidance published since we conducted our trial adequately reflects the available guidance on exercise prescription. We have referred to this in the Discussion.	Reference 17 L417-420
What do the authors mean by 'benefit from additional input'?	Clarified in text to indicate additional to the self-directed rehab approach recommended in the recent NICE guidance.	L459

VERSION 2 – REVIEW

REVIEWER	Cameron, Claire University of Otago, Dunedin School of Medicine
REVIEW RETURNED	03-Aug-2021

GENERAL COMMENTS	Thank you for the considered answers to my questions. However, I still have some concerns. I found the responses to my questions were not very clear in and of themselves. Part of this is the reliance on the published analysis plan without sufficient detail in the response. You talk about statistical significance in response to my query about your reliance on statistical significance which the CONSORT statement does not support.
--

	The study has been carried out appropriately. The fact that you do not have a no treatment control group is not a limitation. It is not possible or ethical to make a comparison with such a group.
REVIEWER	Ganesh, Shankar Composite Regional Center for Persons with Disabilities, Physiotherapy
REVIEW RETURNED	01-Aug-2021
GENERAL COMMENTS	The manuscript reads well and the authors have provided satisfactory answers to the points raised in the first review.